# Health, Security and Fire Safety Process Optimisation Using Intelligence at the Edge

**DOI:** 10.3390/s22218143

**Published:** 2022-10-24

**Authors:** Ollencio D’Souza, Subhas Chandra Mukhopadhyay, Michael Sheng

**Affiliations:** 1School of Engineering, Faculty of Science and Engineering, North Ryde Campus, Macquarie University, Sydney, NSW 2109, Australia; 2Department of Computing, Macquarie University, Sydney, NSW 2109, Australia

**Keywords:** TinyML, machine learning, edge analytics, energy harvesting, health care, security, safety, fire safety

## Abstract

The proliferation of sensors to capture parametric measures or event data over a myriad of networking topologies is growing exponentially to improve our daily lives. Large amounts of data must be shared on constrained network infrastructure, increasing delays and loss of valuable real-time information. Our research presents a solution for the health, security, safety, and fire domains to obtain temporally synchronous, credible and high-resolution data from sensors to maintain the temporal hierarchy of reported events. We developed a multisensor fusion framework with energy conservation via domain-specific “wake up” triggers that turn on low-power model-driven microcontrollers using machine learning (TinyML) models. We investigated optimisation techniques using anomaly detection modes to deliver real-time insights in demanding life-saving situations. Using energy-efficient methods to analyse sensor data at the point of creation, we facilitated a pathway to provide sensor customisation at the “edge”, where and when it is most needed. We present the application and generalised results in a real-life health care scenario and explain its application and benefits in other named researched domains.

## 1. Introduction

The prediction for sensor usage indicates that [1] several billion devices will be installed by 2025, leading to an exponential growth in sensor data [2]. Individual systems in the wild compete to transfer sensor data to centralised repositories, such as cloud infrastructure, for analysis. Communication networks such as LPWAN, 5G, NB/10T or even the WiFi 2.4G or 5Gig (Free To Air) options will find it challenging to maintain the synchronous nature of time-sensitive data with sufficient temporal accuracy that life-saving domains require. The volume of disparate data trying to share network resources progressively reduces the bandwidth available to each user. Analysts also have to manage massive amounts of error-prone sensor data reaching them, making processing and analysis a resource-hungry task that must be conducted efficiently and effectively [3].

Real-time domain sensors are always active [4], monitoring parameters continuously to expose problems [5]. The high detection throughput needs advanced techniques to conserve energy and optimise real-time operational reliability over long, sustained periods. Evaluating the contribution of each sensor type is necessary to determine the right mix of data required in each operating domain. A tabular listing of domain features, Table 1, presents the commonality of needs in the health, safety, security and fire safety domains.

### 1.1. Key Issues to Be Considered

Reports suggest that only snapshots of the generated real-time data are sent successfully due to network constraints and reduced sampling rates. This truncated data, Figure 1, affects the accuracy of research and analysis in real-time operational domains [5]. The raw source data are not sent over networks to reduce privacy risk, communication bottlenecks, costs, processing time, the expense of handling, storing and maintaining infrastructure and data manipulation required by large, stored data sets before analysis.

**Figure 1 sensors-22-08143-f001:**
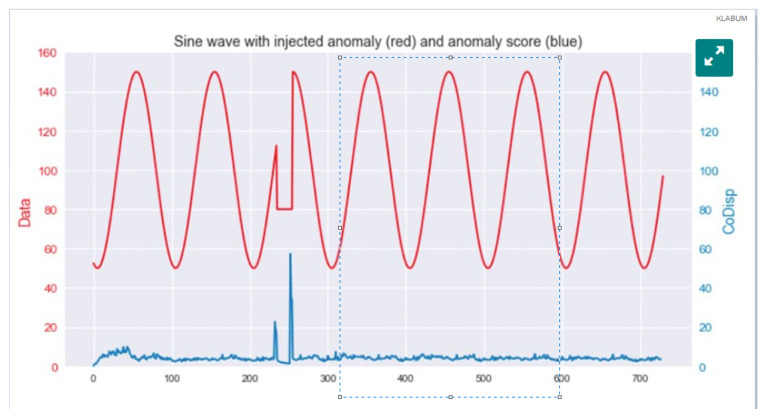
“When time is of the essence” [6] waveform sampling study of truncated data.

### 1.2. Research Considerations

We identified, measured, corrected and compensated for issues that maintain data integrity and operational consistency [7]. Attention to the selection of sensors, the authenticity of real-time data [8], and the fusion of sensor data close to where the data are created is a crucial process adopted. For a pre-defined “wake up” period, constrained low-power computing devices are needed to analyse all the data in real-time to deliver the results to a localised repository. This technique ensures that all essential data are captured and analysed in real time. Machine learning models were created by training them to meet domain operational needs. The post-fusion analysis models developed in the cloud were distilled down to versions using TinyML (a compressed version of the larger version) to fit onto constrained microcontroller platforms, as presented in later sections. Results are sent as an optimised “payload” to a “first responder” [9] or a central control facility [10], effectively reducing the requirement for temporally synchronous computing resources [11] at the management end.

Conserving power is essential in field-deployed devices, so domain-specific anomaly detection “wake up” triggers turn on power-hungry analytic modes to optimise the power consumption of the microcontroller devices.

In health care, data from sensors have an essential role in maintaining operational care compliance and improving the response to those that need urgent care [12,13]. These deliverables are stipulated in legislation; hence, measuring and sustaining performance in real-time is a serious requirement.

In the security, safety and fire safety domains, the need is to generate an alert based on a multifactor assessment of complex sensor data, to keep “unwanted event” triggers to a minimum. Industry statistics for false or “unwanted” alarms are close to 90% [14], resulting in a waste of resources, time to attend these events and severe economic loss in some cases. In all these situations, constant changes occur, and sensor management systems that send ineffective post-event advice continue to disrupt normal processes [15]. Our method improves detection quality and ensures event predictability to improve outcomes [16]. Flow charts, tables and a framework based on machine learning techniques at the “edge” explain the model’s detection, prediction processes and performance outcomes.

Highlights and original ideas over existing state of the art, presented in several sections of the paper, are summarised for clarity. The basic idea of using machine learning is not new, but the creation of a model to overcome specific “real-time” industry domain application issues is one of the key highlights. Our development overcomes issues with the management of real-time operational process flows and optimises situational awareness by delivering event information directly to the “first responder”. Several examples demonstrated in different sections show how “event detection” is more than just the result of a single sensor trigger but the result of a “cluster” of sensors of different types delivering “intelligence” using “collaborative, corroborative and reinforcement” techniques. This, we believe, is our unique contribution using machine learning in health, safety, security and fire safety where time is of the essence and critical to life-saving processes.

## 2. Materials and Methods

Sensor data from monitored parameters at the site [17] are part of the organisation’s established risk management strategy. Identifying and managing data loss, data “cleaning” and data storage in the repository from where the analytic engine ingests, corrects and analyses data to comply with the requirement of the data-driven environment is a core responsibility [7]. The two key issues that reduce the efficacy of many data-harvesting operations are the quality of data arriving and data analysis in temporal sync. The first issue tackled is the quality of data. Multiple sensors deliver multiple data streams, so data fusion techniques reduce a sensor cluster’s results payload. The results are sent directly to the “first responder” dashboards to improve real-time situational awareness (see simplified explanation in Appendix A).

### 2.1. Related Work—A Two-Year Quest to Meet Real-Time Smart City Objectives

A prior two-year design and implementation of a smart city research system funded by local government and a university research endowment provided adequate experience in data management. The knowledge and understanding of how vital temporal sync is to support health and well-being concerns are documented in system configuration drawings (graphs in Figure 2) [7], data management records, published papers and reports. The research explained how seventy-one in-house developed multisensor IoT nodes connected to two city-wide gateways delivered vital data over public LoRaWAN infrastructure. Over two years, the experience demonstrated that reliability, data throughput and temporal sync were vital to the usability, effectiveness and efficiency of city-wide services. Other instances also reinforce this conclusion [18].

### 2.2. “Intelligence”—Training Machine Learning Resources to Produce Efficient and Effective Outcomes

“Intelligence” describes the outcome of a fusion process used to securely package the results of the data analysis at the “edge”. This process no longer requires expensive data handling to produce accurate results or to maintain the integrity of the data at the point of collection where the data generation and analysis will occur. Results from analytics at the “edge” are fed directly to a live operational dashboard [19].

The process uses techniques that include the use of “situational knowledge” or “features” (Table 1) essential to train the machine learning system to improve decision-making and prediction capability. Machine learning (ML) [20] requires a training data set to extract associations and insights from disparate data. Hardware-constrained microcontroller “edge” devices use (TinyML) [11] models created in the cloud to function at the “edge” [21]. These models enable sensor fusion analysis to improve “classification” and detect anomalies, leading to an improved detection performance and prediction capability.

### 2.3. “Feature” Logic Designs for Targeted Real-Time Domains

The sensor grouping enables the mapping of multiple sensor outputs to “collaborate, corroborate and reinforce” to confirm an event. The concept developed in this research is explained diagrammatically in Figure 3. The human algorithm builder can only do so much based on the associations seen in the data. Machines can do this many times faster and build many more unbiased associations. The core learning is that allowing ML to extract relationships, associations and relevance produces better insights but requires plenty of relevant data to be used in training, as per the logic flow example described in Figure 3.

### 2.4. “Event Reinforcement” Techniques Using “Collateral” Sensor Data Example

The process workflow of a “reinforced” data point is explained in the flow chart Figure 4 and Table 1. The accompanying “pressure wave” is detected if a properly shut door is opened. This “collateral” sense confirms the door has been opened from a shut position rather than from an open (ajar) state.

A similar phenomenon is used to “corroborate” activity such as using “gesture” recognition. Just the motion sensor triggering might not help define the event, but a side-to-side motion (directional detection) would generate a unique motion signature as temporally synchronised “collateral” data from the motion sensor on board. This event could be assigned to a different response or output—such as a “nurse call” to care staff.

### 2.5. Visual Colour Sense “Camera”-Like “Features” (No Privacy Broken—Single-Pixel Resolution)

The “reinforcement” techniques Figure 5 are essential because the ability of the sensor to detect colour, albeit just a few pixels, “reinforces” sensor data that point towards a specific characteristic such as the colour yellow in fire or red on a caregiver uniform.

Humans build unique algorithms with the capability to build associations using a few sensors. Machine learning can build associations using many or all sensor inputs of different types and in temporal sync. This widens the opportunity to find similarities, correlations, dependencies, links, etc., to enrich the data and detection quality, accuracy, depth, reach, etc., and perform it in seconds to ensure that nothing is missed.

In the case shown in Figure 5, a single pixel colour detection sensor is already on board. Therefore, associating this colour detection with the colour of the nurse uniform will reinforce the association of detecting caregiver movement near patients. In addition, in the case of a fire, the yellow colour reinforces the existence of a flame.

The few associations developed by humans are helpful but not as useful as the thousands that are possible across the entire operation, which machines (MCU and ML) can perform very fast in human time.

### 2.6. The Feature Set and Performance Specification

Key system operational “features” and performance requirements are shown in Table 1.

### 2.7. Machine Learning Training Requirements and Operational Examples

The targeted domain-sensitive training outcomes are shown in Table 2. The domains considered are a whole class of environments that require real-time event detection and immediate response. The physical environment is not so critical, for example, indoor, outdoor, small room, large hall, bedroom, lift, etc. The key issue is “what is to be detected?” (i.e., the objective). Enough examples of the type of event are required in the training of the ML model. ML will build associations based on triggers from various sensors in the systems that are in temporal sync.

### 2.8. Examples to Explain How ML Would Build Associations, Collate, Collaborate, Corroborate, Reinforce Better Than Humans by Using Multiple Sensors on Board the MCU

A motion is detected along with a “blue uniform” colour recognition trigger; it does not matter where the incident occurs. The association of two key known criteria, person movement and the colour of the uniform worn by the responder, would flag a caregiving episode. Similarly, a vehicle travelling outdoors and a group of people detected in the path could be an accident waiting to happen. In a room, if a door was opened and the pressure sensor was also triggered, then the association of the opening door and pressure wave indicates the opening of a closed door, not a door left ajar. Some other sensors will indicate human movement, which could mean intrusion, the arrival of help, etc. ML thrives on building associations of such value that detection of incidents (with recognition of event type) or “anomalies” (unusual activity) is no longer conducted by a single sensor but by a group of sensor types because of the speed and capability to create associations by a trained ML model. This heightened situational awareness is due to the creation of many associations made by the ML model during training. This becomes an efficient and effective tool to support a human operator’s “inference” capability. These benefits are explained in the decision tree tables and workflow diagrams provided later in the description of the trials.

### 2.9. Implementation of an Experimental Environment to Test the Hypothesis

At the “edge”, each domain will have a “wake” trigger to activate the “intelligent edge analysis” on the data generated by a sensor or a group of sensors (fusion) [22]. We put together the target “behaviour” outcomes for our training set, as listed in the domain segmentation Table 2.

A small system combines location- or task-specific sensors [7]. A cluster of small systems in a larger configuration (i.e., multiple small systems tied together) require a federated learning approach to using “intelligence” at the “edge”. We propose scalability using a fusion sensor approach and BLE (Bluetooth Low Energy) communications.

### 2.10. Applied Traditional and Machine Learning Workflows

We use value engineering [23] to design the model to ensure it is scalable. To this end, all logic (including design, programming and simulation) goes into delivering these benefits Figure 7.

Live simulations, from the existing system, with as many examples of match and non-match as possible, were used to meet machine learning (ML) ingestion workflow standards (Figure 8). The accuracy of the model generated by the machine learning build process depends very much on the variations fed during training. After this training, the model serves as the “selector”, looking through all the new data to explore all associations, relationships and relevance to produce the best match determination that improves situational awareness.

### 2.11. Embedded Systems Evolution and Deployment in This Research

Embedded systems with I/O and processing power deployed at “the edge” is enhanced using BLE (low-energy mesh communications). Typical system configuration [24] is shown in Figure 9. Tiny Machine Learning (TinyML) reduces the size of the generated models to fit into resource-constrained hardware with BLE capability enabling an ecosystem to emerge from this work. Each “cluster” of sensors on board an MCU device forms an intelligent node. As we expand connectivity between “clusters” over BLE, the network grows in capability to share information [25], and “intelligence at the edge” is created.

### 2.12. Architecture Deployed and Lessons Learned

Multisensor microcontroller (MCU) boards currently have physical, chemical, environmental, video, audio and optical sensors built on the single embedded board. These multidimensional data sensor architectures provide machine learning systems access to rich data to unearth observations that contribute to a greater understanding of each dimension’s inputs to the system’s overall operational performance.

As demonstrated earlier in logic diagrams, ML-based event detection is no longer a single sensor endeavour. ML uses thousands of associations to collaborate, corroborate and reinforce new scenario settings and unique event detection episodes.

The MCU used is the Arduino Nano BLE Sense with eleven (11) independent sensory capabilities in five (5) sensor groups. These MEMS-based sensors are very reliable, capable devices on the MCU as a cluster of sensors of different types—ideal for machine learning to generate complex associations (Figure 10).

Our research uses off-the-shelf MCUs. The choice, capability and sensor density vary; hence, as a first step, sensors relevant to the domains were identified. In the health, safety, security and fire safety domains, person detection, parameter detection (such as temperature, humidity, air quality, airflow, hygrometric levels wetness), vibration, movement, colour, pressure and sound thresholds (unique sounds and keywords) were prime requirements.

Applications in health care are explained with bed-mounted and wall-mounted configurations shown later in graphed results. Each MCU with sensors on board act as a “sensor cluster”. They have an ML model uploaded to it and perform as per the training provided during the model creation. Layouts and images of actual test training procedures and set ups are explained with relevant test frame grabs.

There is no single sensor per area because each MCU is a sensor cluster connected to other clusters via BLE. Each cluster could have eleven or more parametric sensory outputs to an ML model, which would then proceed to extract events, anomalies, etc., based on the model uploaded to it. By using associations, it generates from multiple sensors as explained in the collaborations, corroboration and reinforcement framework workflows; ML conducts them many times faster to present a range of associations and result options to a live dashboard.

TinyML models are uploaded to different form factor MCU configurations. Tiny compute hardware, with constrained resources and different onboard sensors, as shown in Figure 11, provide a rich source of functional performance measurements required in health, safety, security and fire safety environments. Thermal sensor add-ons for predictive fire safety also follow the same principles but use an extended spectral dimension to predict the onset of fire risk.

A systems approach establishes steady-state operating parameters. This process is shown in graphs produced by device simulations in fixed and flexible configurations. It measures quiescent operating conditions and real-time sensing variations for different sensor inputs, which provides an understanding of how the device (and ML model) would function in a fixed or tailored configuration. Most of these simulations use cloud services connected live to the “cluster” or use data uploaded to it, which is usually the process followed to “fine tune” the system.

### 2.13. Machine Learning (TinyML) Sensor Fusion Process

The fusion of all relevant detection input data is performed by uploading all relevant data to the service provider cloud repository. Once all data types are uploaded to the cloud after all preliminary sensor tests are finalised, the cloud service creates the model in the cloud using the software learning block apps provided. Subsequently, the model is reduced to run on a standard low-cost microcontroller (MCU) [22] device installed, for example, on a wall (Figure 11), with the TinyML model on it. The MCU and model run at less than 1 mW in “listening mode” (awaiting the “wake word”). When triggered, they take up to a maximum of 10 mW to process an event and then drop back into “listen” mode (Figure 12).

One of the many industry-developed cloud-based tools was used to understand the process and work on the key steps in producing an “intelligent edge device”. Current device technology uses “digitised” (MEMS [26]) sensor technology, so a combination of sensors are built on board (Figure 12), allowing the “edge intelligence”application (Figure 13) that performs all existing functions to be implemented.

Site layout for BLE mesh communications.

### 2.14. Deploying a Machine Learning Process

The “tooling” is a term used to describe the set of instructions, methods, support software and infrastructure provided by some providers of cloud-based ecosystems to build TinyML models. Support infrastructure allows uploading sensor data and external sources to the cloud for analysis and preparation [27,28], before the model is created.

Figure 14 describes the workflow followed when developing the solution from live data [5], which are collected based on training requirements to ensure the trained model delivers effective outcomes. Sometimes the results are below expectations, so retraining is needed, or the results are very encouraging and can be installed confidently.

### 2.15. Training the ML Model (Figure 14)

The key steps to follow in planning the application in the selected domains are:Choosing the right MCU “appliance” (sometimes referred to as the microcontroller or “silicon”).Collecting the data (achieved using the device itself connected to the cloud service provider). This data can also be collected locally and uploaded to the cloud service storage for analysis.

The service provider also provides the testing and training tools that help create a working model in the cloud, which one can test, verify and test again on unseen data. After the successful first passes based on the domain requirements, the next step is to “squeeze” the model down to the device requirements and upload it to the firmware. At this point, a fully functioning “intelligent edge device” emerges with a model loaded to its firmware customised to suit our application. The Actual Working Steps (soft considerations in the application such as the “Help Me!” call) are expanded below. The steps to successfully create a model in the cloud and upload it to a microcontroller device vary from provider to provider. Details on how we achieved this follow.

### 2.16. The Application Objective

Choosing the ecosystem described earlier, we needed an “intelligent edge” device to meet the detection requirements of four distinct domains with very similar detection demands (see Table 1 and Table 2).

### 2.17. Data Collection—Designing What and How

The microcontroller device with multiple sensors on board is connected to the service provider server via a custom CLI or a WEBLink. Figure 15 demonstrates the process of collecting data for the “wake word” “Help Me!”.

The system designer makes a distinct choice on what the “wake word“ will be. “Help Me!” is a common but distinct two-syllable phrase. The training outcomes in Figure 16 show that 95.4% accuracy was achieved.

### 2.18. Collecting and Obtaining the Right Data

A comprehensive data set was collected from multiple sensors to help the ML process find many more associations between sensor triggers. The “wake word” will trigger the analytic system to respond to a higher-level risk analysis triggered by the microphone sensor, thereby conserving energy to process the other “feature detection” requirements designed into the ML application. We found this to be a powerful framework that allows the ML analysis tool to optimise the associations using the “collaborative, corroborative and reinforcement” techniques designed into the framework. The ML process will find many more associations in temporal sync, depending on the level of training data used, which is just impossible using human observation alone, as shown in Figure 17 below.

Process flows presented earlier show “collateral”, “collaborative” and “reinforcement” data analysis techniques confirmed the occurrence of “events”, reducing the “FALSE ALARM” rate that beleaguers the industry. Machine learning analysis produces many more associations when using the framework’s “collaborative, corroborative and reinforcement” techniques. Learning to develop the model in the cloud with adequate computing resources is a very effective method to develop the “wake word” mechanism from scratch.

### 2.19. Summary of Materials and Methods

Our research exposed several areas of interest in the quest to build a universal sensor for the chosen domains. We focused on health care and its immediate needs, i.e., to respond to issues of care.

To call for help assistance/support, one “wake up” call, “Help Me!”, was developed. Once this “wake word” is invoked the system begins to analyse issues based on the training objectives of the TinyML model.

The bed vibrations with sensor fixed to the wall Figure 18 and bed vibrations with strapped sensor Figure 19 directly triggers a call for help (as just another “wake up” trigger) with better precision than a human who might pick up only one aspect of the problem. Most occurrences can be “taught” to the ML system using pre-recorded event history. Sensor data play an essential part during collection to provide input on the temporally synchronous activity that can precisely identify the problem.

Audio is used as a “wake word” to provide the best opportunity for the patient to be heard. Training ML on how to ignore irrelevant sounds was conducted as explained in the following graphs. The strapping proved to help “listening” because all sympathetic noise generation is smoothed out with the strapping (Figure 20 and Figure 21).

The analysis of the “anomaly” triggers from multiple sensors determines much more than bed shaking. The multidimensional data from, for example, the magnetometer, can provide information on the bed itself being moved, misaligned, etc., leading caregivers to investigate why (Figure 22). In all domains the scope of this technology is very encouraging.

Failure cost is a serious consideration because operational performance depends on reliable sensors with long battery life. We considered low energy management strategies such as the “wake word” and BLE (Bluetooth Low Energy) connectivity and are investigating energy harvesting options such as infrared wireless power to extend the system’s operational life.

### 2.20. Application Overview

Health, security, safety and fire safety are the four domains targeted in this research (Table 2). The configurations were developed to use a multidimensional approach using MCUs with multiple “digital” sensors on board to maximise the sensor fusion outcomes. Fusion techniques enabled ML to optimise the benefits of having many data points to collaborate, corroborate and reinforce outcomes.

The BLE [29] device uses several modes to communicate with other devices or the outside world (even via the internet). These mechanisms follow the Generic Access Profile (GAP) guidelines. GAP defines how BLE-enabled devices can make themselves available and how two devices can communicate directly (Figure 23). The technology is flexible, reliable and has a maximum range of up to 1 Km with a suitable antenna.

### 2.21. Data Used in Training the Machine Learning Model

Data were generated from two sources:(a)Simulations (because site access was not possible for some tests).(b)Anonymised data provided from actual live sites.(c)The simulated data collection described in later sections.

The anonymised data were generously provided by a trusted external source and contain the following event information:Access control data describing entry and egress events in temporal sync.“Nurse call” data describe health care residents’ activity in calling for help using the push button on the bed.

### 2.22. Using Data Provided along with Simulated Data to Train Model

The process of using different data structures is quite complex but made easy using tools provided by the TinyML service provider (Figure 24).

The data samples are shown in Figure 25 and Figure 26.

This data had to be converted into a format that is time stamped and follows the prescribed format: timestamp, Data-1, Data-1, Data-3, ….

The data (Figure 25 and Figure 26) were uploaded using the interface provided (Figure 24). The ML model creation was initiated in the cloud service. Advanced “learning blocks” (sophisticated processing steps Figure 27) such as Transfer Learning and Regression filters, NN, etc., were provided to improve model performance.

The TinyML ecosphere is still developing, allowing many opportunities to discover new techniques to improve the performance of models. The key outcome in this research is understanding the stepped approach to developing models that make it possible to have “intelligence at the edge” embedded into constrained computing devices such as microcontrollers.

## 3. Results

### 3.1. “Feature”/Function Table

The results table explains how different aspects of the project objectives were met with system flexibility and performance in focus.
**“Feature”/Function****“Edge Intelligence” Using Sensor “Clusters” and Low-Energy Techniques**Quality of sensors and functionalityMany unique MEMS low-power reliable sensors on computing device board (five parametric sensing groups, eleven sensor types).Sensor fusion in real-timeSensors are in close proximity, close to where the data are captured, so optimised, temporally. Synchronised data fusion analysis is successfully performed.Data collectionRelevant data in many forms can be added to the fusion step.Power optimisationNew techniques such as “wake words” conserve power for those times that require computing resources.Advanced feature setsData are analysed in real-time—only results stored or sent to storage or dashboards. The result “payload” is in Kilobytes. Machine learning extracts associations from the data to improve situational awareness. It can achieve this many times faster than humans can after appropriate model training. “Intelligence”The power of associations generated by ML enabled “collaboration, corroboration and reinforcement” avoiding “unwanted” “event detection” based on just a single sensor trigger. 

### 3.2. The “Wake Word” for Energy Harvesting

Activating and testing the “wake word” simulation returned a 95.4% accuracy rate (Figure 28). Tests were conducted on the cloud service of EdgeImpule.com (personal demo account).

### 3.3. The Confusion Matrix

A table of the results using the confusion matrix for event detection using machine learning is recorded in Excel. The number of tests and trials required automatically calculating the results of the “model testing” in an automated spreadsheet, made it easier to record the consistency and to test different scenarios providing the flexibility to try out many different operational modes: voice, time sync data, fusion techniques, etc.

The trials (Figure 29) were carried out to experience the machine learning process and to experience its benefits, applicability and how it relates to existing processes. We also felt that the real power of TinyML is its ability to deliver a model which can be improved at will with training and is uploaded to a microcontroller device to perform application tailored analysis.

#### Automation to Test Multiple Scenarios

A spreadsheet (Figure 30) was developed to automatically take in data and produce confusion matrix results so many tests and trials could be conducted for comparison over multiple tests.

## 4. Discussion

The most effective application of model-based sensor fusion would be in a smart city project. We juxtapose two scenarios and represent them in drawings and graphs.

Scenario-1 is a smart city project with 71 sensor nodes sending data over a LoRaWAN network to a central repository. In the design requirements there were no real-time expectations. Therefore, the system performed as per the resources provided to it. Independent sensors were used, which shared a LoRaWAN communication channel and produced Max1 to 100 Kbits/sec of data to a repository. The data were analysed for insights, then transitioned via an analysis package to a dashboard several minutes later depending on network traffic delays and retries or loss and “cleaning” functions to replace lost data. The time to collect, clean and have data ready for analysis for a real-time threat response was in the order of ten to twenty minutes (Figure 31).

The second test scenario was created using the Arduino Nano BLE Sense with many more sensors on board. After ML training, presented earlier, the system detected and performed instantly over a sub-giga-hertz BLE (Bluetooth Low Energy) channel providing “actionable intelligence”, i.e., human deliverable responses.

From the two studies, the following scenarios are described:(a)Individual sensor data are sent over severely under-resourced networks to a central repository for processing, where issues such as network availability, speed and reliability are not guaranteed. Background analysis and “cleaning” (i.e., replacing missing data) at the receiving end is required before the results are sent to a dashboard as events for operators to respond to (Figure 31).(b)Another scenario where “intelligence at the edge” generates analysed results using a customised machine learning model where the data are fused and analysed at the point where the data are collected, and “actionable intelligence” is sent out to the first responders (Figure 32).

These two scenarios present two different ways of delivering “actionable intelligence”, which needs to be “on time” or it is of no benefit to the end user.

### Limitations

The “need for speed” to deliver solutions that work forces humans to take short cuts, which does not sit well with machine learning because it expects us to know what kind of an output we want. “Garbage in–Garbage out” is so very true of the machine learning process that works incredibly fast and accurately. The process workflow was explained in depth, but if it is “compromised” by poor input or “reasoning” it is likely to magnify the lack of rigour in the assumptions made and outcomes delivered.

## 5. Conclusions

Our research explored, experienced, documented and presented several levels of discipline required to obtain reliable results when configuring machine learning systems. We researched and reported preliminary results from the domains where our TinyML model’s customised detection capability was trialled, documenting the “features” and model functionality outcomes required, as shown in Table 1 and Table 2.

An energy-efficient single-board microcontroller system with MEMS sensors on board (Figure 10) and BLE communications was set up in a test scenario. As the MCUs (embedded microprocessor control units) develop in sensor density and computing power, the opportunity to use low-power, high-throughput “meshed” (BLE inter-device communication) devices can only grow to improve “edge intelligence”. This area is highlighted for future work.

Our research explored and deployed these sophisticated customised sensor models using data fusion and techniques that continue to be researched in the industry. The process of training, identifying features and functions that need to be used in training, is an expertise that must be practised. We tested and delivered customisable TinyML model designs to meet real-time operational domain requirements using “edge intelligence”. These developed models can be used with many different MCUs with similar sensors and in other domains with similar detection requirements. This solves a significant issue for industries with severely restricted resources. We explored all avenues to ensure reliable data analysis at the “edge” delivers credible reliable results in temporal sync. Therefore, the results are received by “first responder” hand-held or control room dashboards so that they can be responded to immediately reducing “unwanted” or false alarms that are a drain on resources.

This research also explored and presented the possible sophisticated framework that uses data points to collaborate, corroborate and reinforce results in a data-driven operational environment using multiple sensor inputs, using four real domain requirements as examples.

Failure cost is a serious consideration because sensors contribute to operational performance and must be highly reliable. Poor reliability will require a human to physically attend to a system failure to find out the problem. Nevertheless, poor-quality products tend to add costs exponentially to the operational cost of a data-driven ecosystem. It is essential, therefore, to use value-engineered design principles to test and deploy with reliability in mind.

Further research into this area of “edge intelligence” is necessary along with key topics covered in this research paper, such as energy harvesting and wireless power, to extend reliability and satisfy value engineering criteria. TinyML is a formidable development and a useful tool for using sophisticated machine learning training methods to improve the performance of the customised models in real-time health, safety, security and fire safety situations.

## Figures and Tables

**Figure 2 sensors-22-08143-f002:**
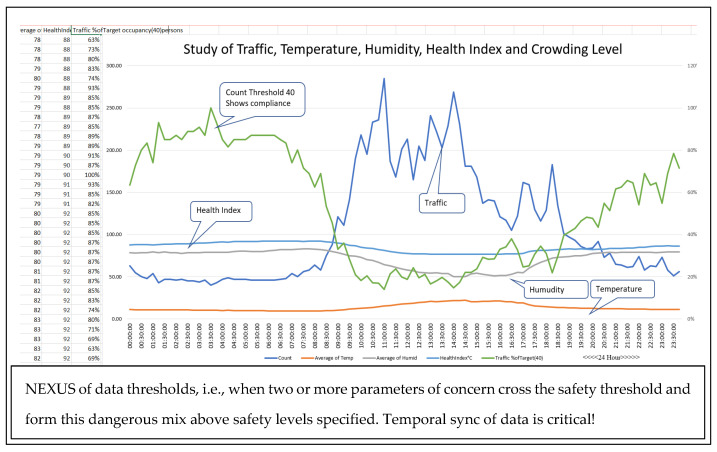
Related prior work building a city-wide IoT infrastructure for health and safety.

**Figure 3 sensors-22-08143-f003:**
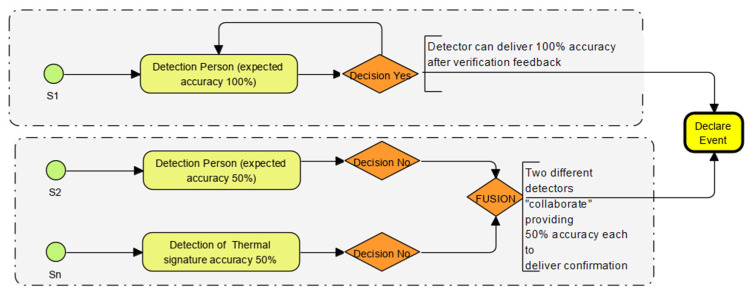
Reinforcing and collateral data workflows demonstrate how data from different sensors “collaborate, corroborate and confirm” outcomes. (Sensors S1 to Sn, Probability & accuracy in yellow, decision points in mauve diamond and outcome from “fusion” in yellow).

**Figure 4 sensors-22-08143-f004:**
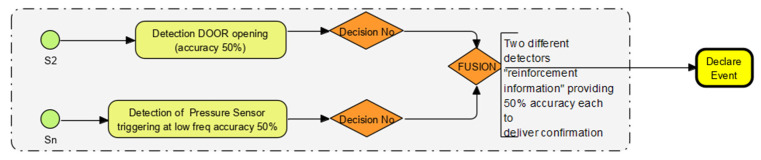
How one sensor data point and another sensor data point “collaborate” and “corroborate” to confirm an event.

**Figure 5 sensors-22-08143-f005:**
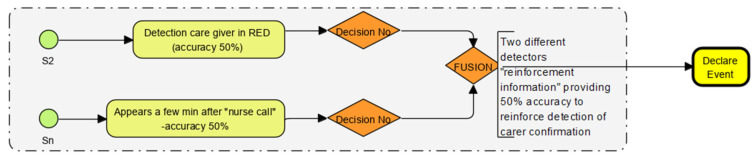
Using colour information to “reinforce” the arrival of a “nurse” in red clothing just after a “nurse-call” trigger is initiated. (sensor data in green detected but not accurate enough at decision point in mauve diamond. However at a fusion decision point in mauve diamond they reinforce and confirm the event in yellow).

**Figure 6 sensors-22-08143-f006:**
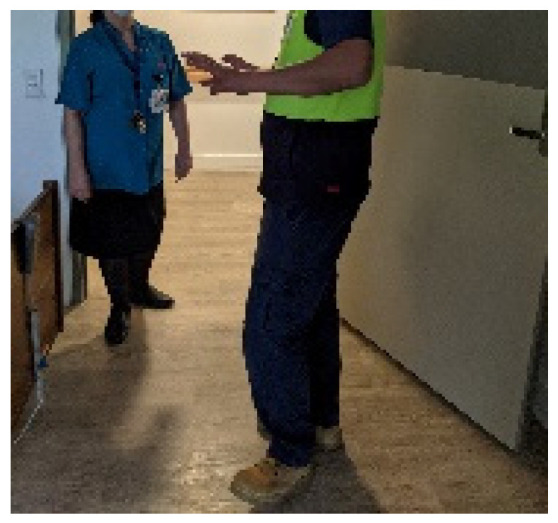
Staff in uniform.

**Figure 7 sensors-22-08143-f007:**
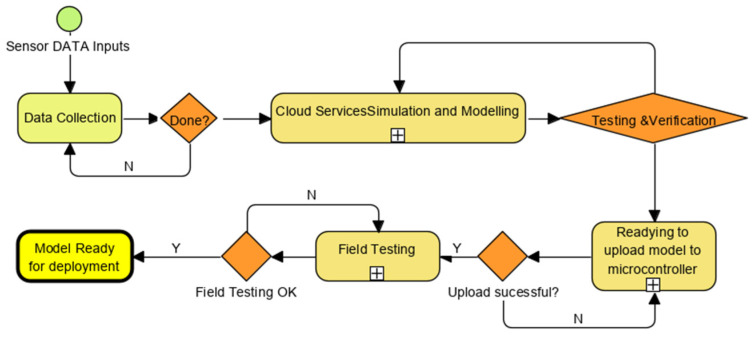
TinyML Process Workflow.

**Figure 8 sensors-22-08143-f008:**
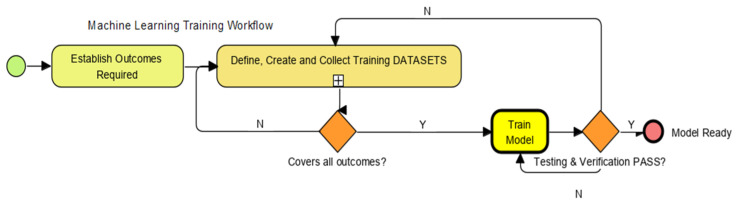
Machine Learning Training Workflow.

**Figure 9 sensors-22-08143-f009:**
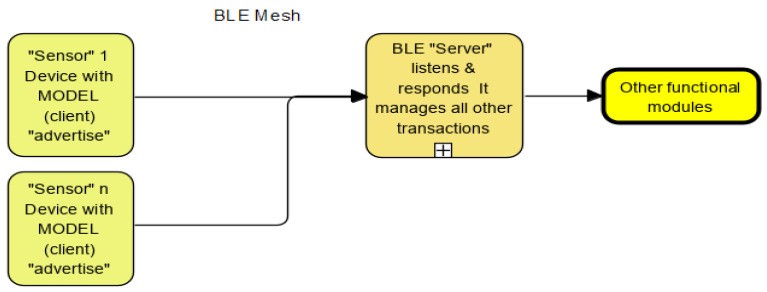
Typical machine learning BLE mesh architecture delivering “intelligence at the edge”.

**Figure 10 sensors-22-08143-f010:**
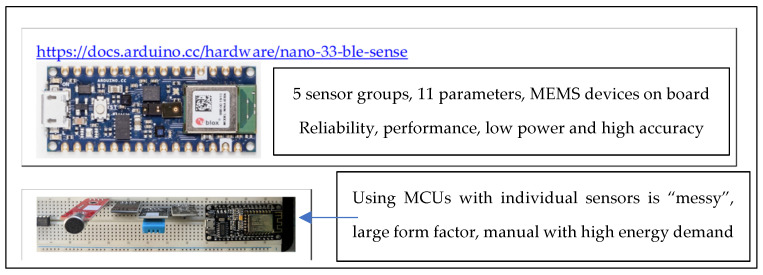
MCU features and choices.

**Figure 11 sensors-22-08143-f011:**
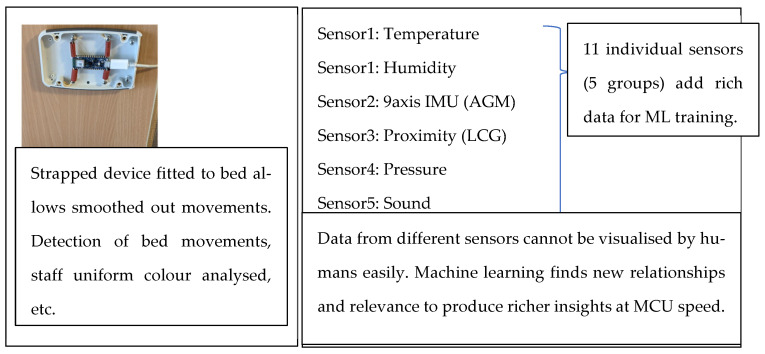
Example of an installed multisensor microcontroller device.

**Figure 12 sensors-22-08143-f012:**
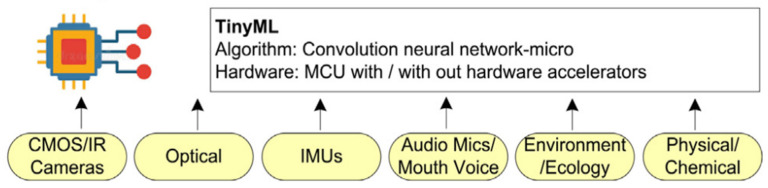
Industry-issued figure of a device ecosystem.

**Figure 13 sensors-22-08143-f013:**
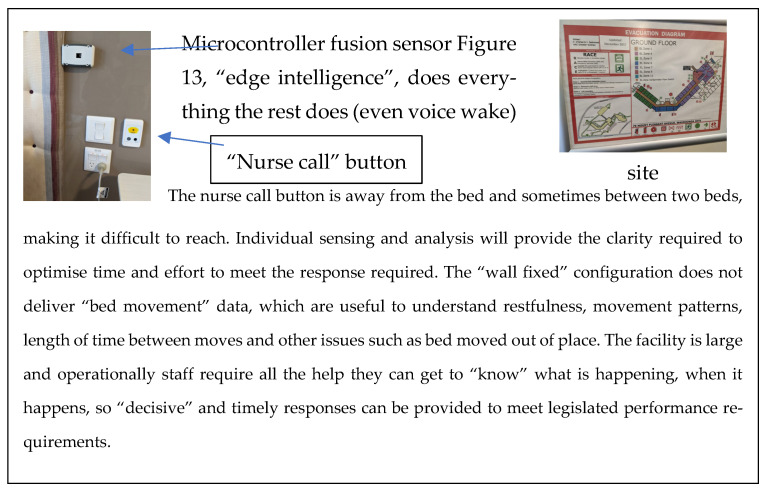
Typical bedside “resident” support assets.

**Figure 14 sensors-22-08143-f014:**
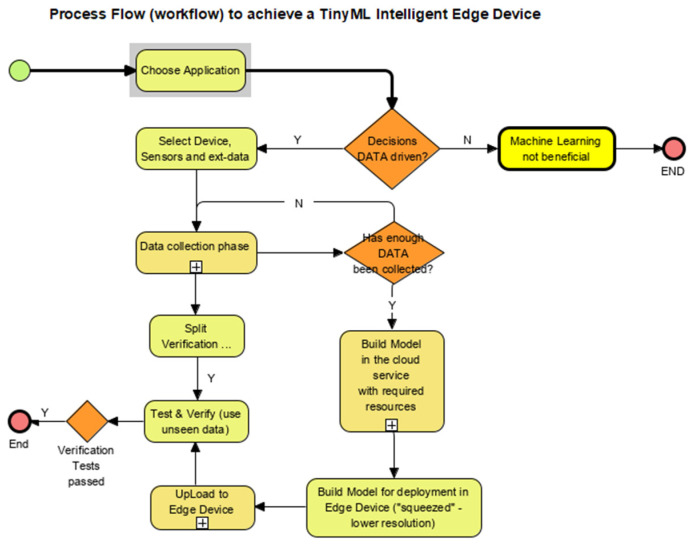
Typical workflow to deploy TinyML.

**Figure 15 sensors-22-08143-f015:**
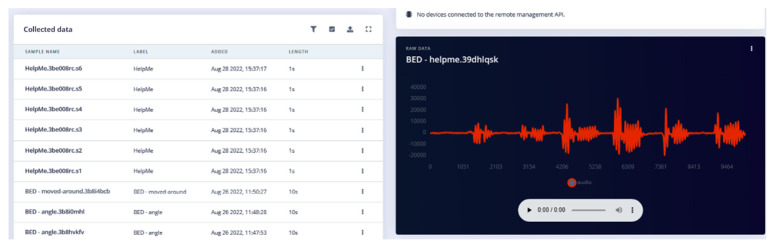
Collecting data for “Help Me!” “wake word”.

**Figure 16 sensors-22-08143-f016:**
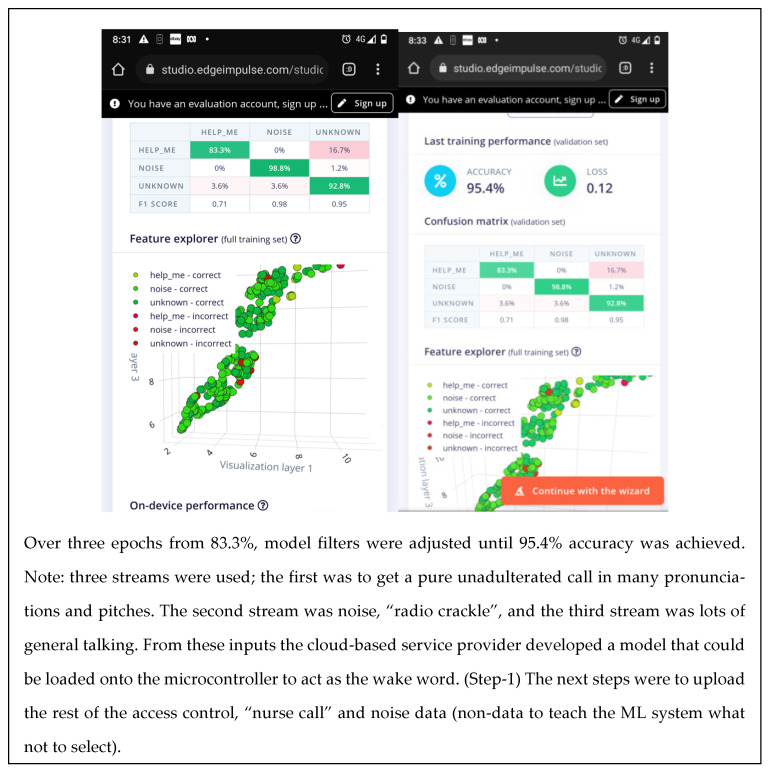
“Wake word” planning, configuration, tests and results.

**Figure 17 sensors-22-08143-f017:**
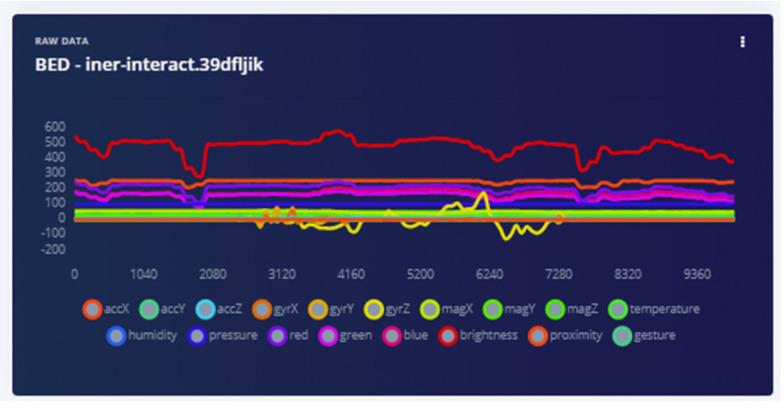
Multiple sensors produce rich data for the ML algorithm to learn and build unique associations within the synchronous time frame (i.e., in temporal sync).

**Figure 18 sensors-22-08143-f018:**
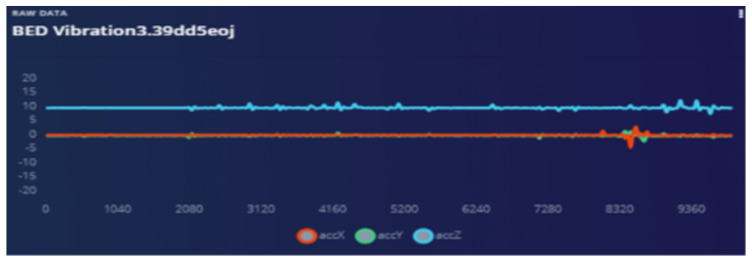
Bed vibration with fixed sensor.

**Figure 19 sensors-22-08143-f019:**
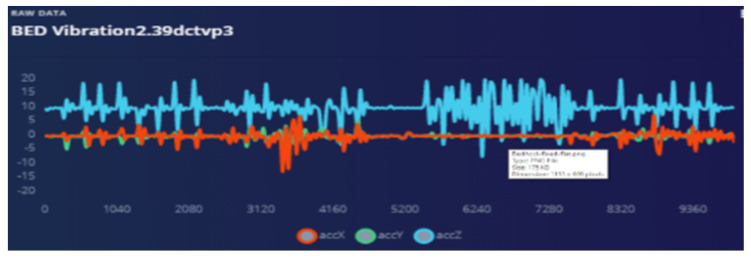
Bed vibrations with strapping suspended sensor.

**Figure 20 sensors-22-08143-f020:**
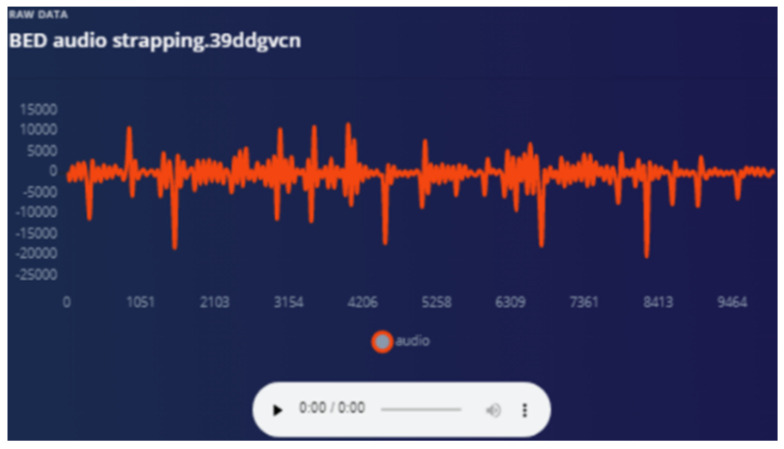
Audio sensor is strapped—waveform smoother.

**Figure 21 sensors-22-08143-f021:**
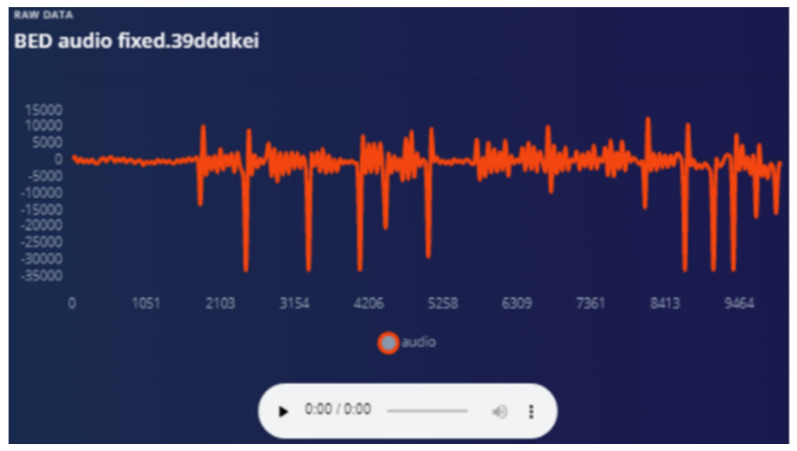
Response from audio when sensor fixed to a hard surface.

**Figure 22 sensors-22-08143-f022:**
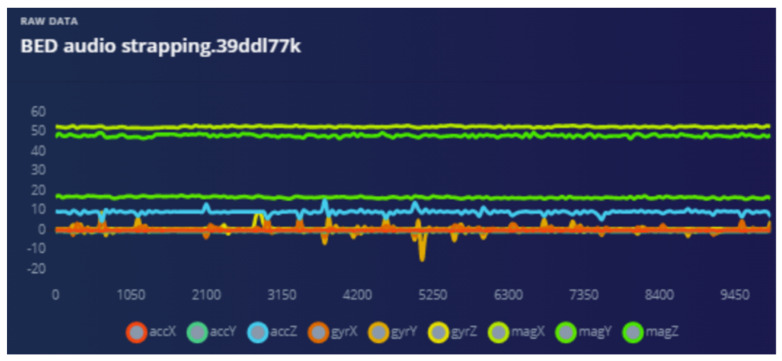
The accelerometer and magnetometer provide positional information instantaneously when disturbed.

**Figure 23 sensors-22-08143-f023:**
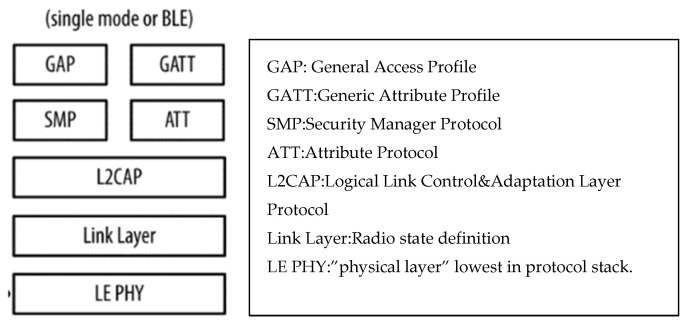
BLE protocol stack that allows meshed and long-range links.

**Figure 24 sensors-22-08143-f024:**
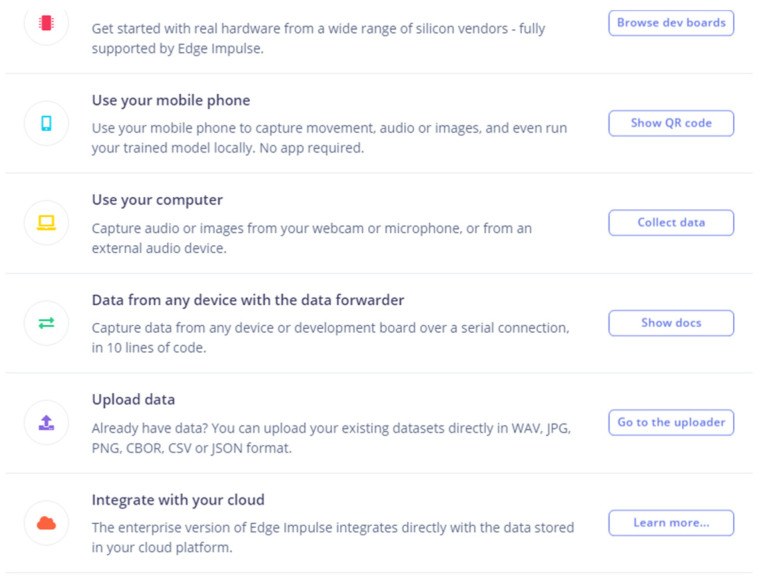
Data upload options provided by the service provider.

**Figure 25 sensors-22-08143-f025:**
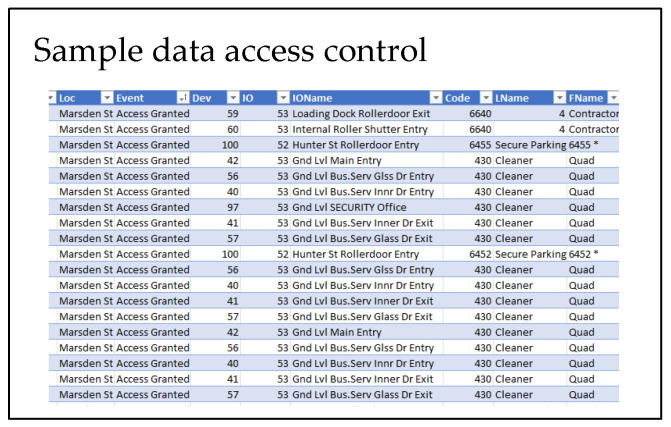
Access control data sample.

**Figure 26 sensors-22-08143-f026:**
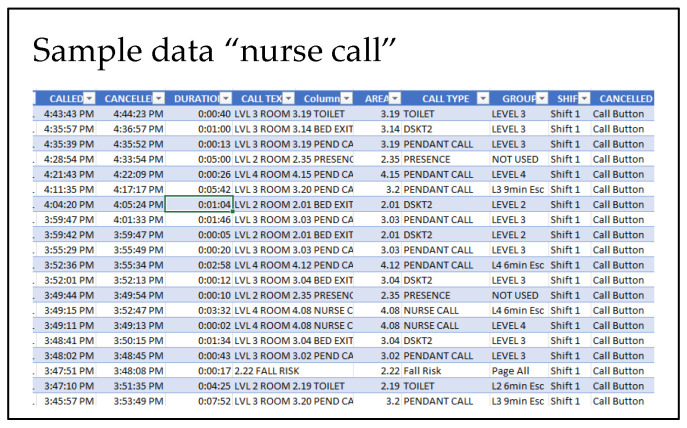
“Nurse call” data sample.

**Figure 27 sensors-22-08143-f027:**
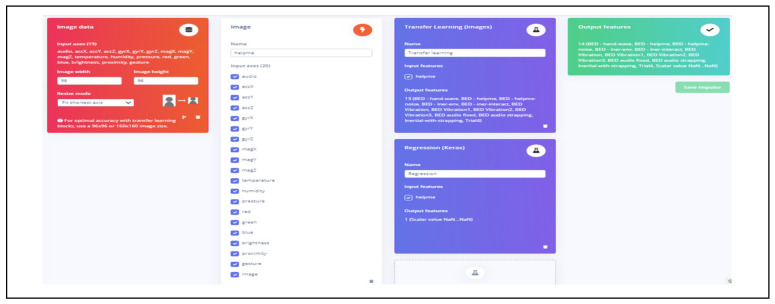
Additional “learning blocks” (TL, KERAS, NN) provided to improve outcomes.

**Figure 28 sensors-22-08143-f028:**
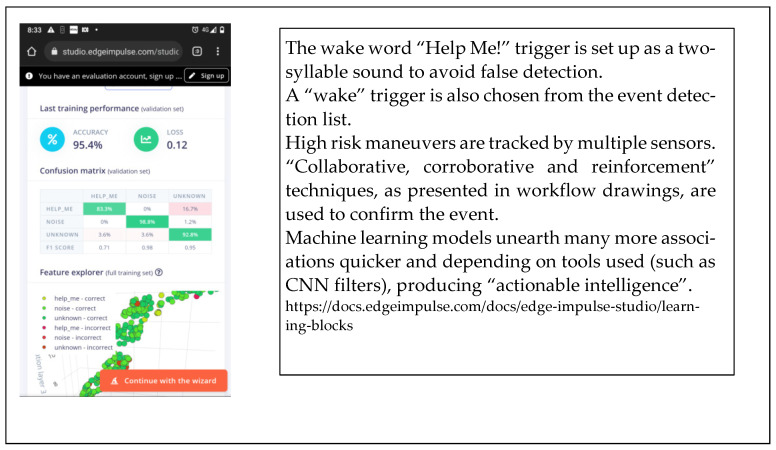
The wake word “Help Me!” turns on event recognition computing resources.

**Figure 29 sensors-22-08143-f029:**
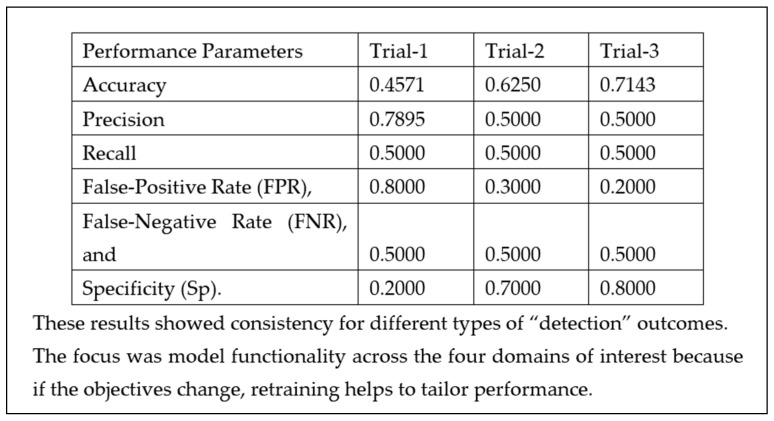
Test results—”event detection” trials.

**Figure 30 sensors-22-08143-f030:**
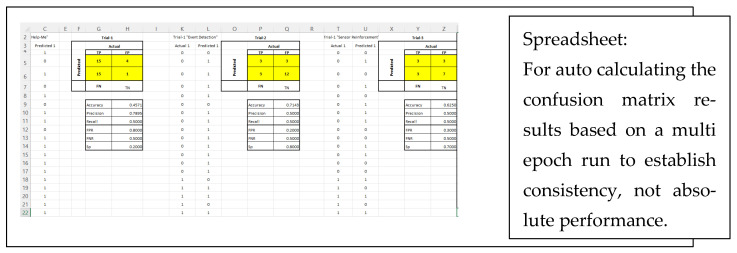
Spreadsheet to perform multiple trials to test for consistency using the confusion matrix.

**Figure 31 sensors-22-08143-f031:**
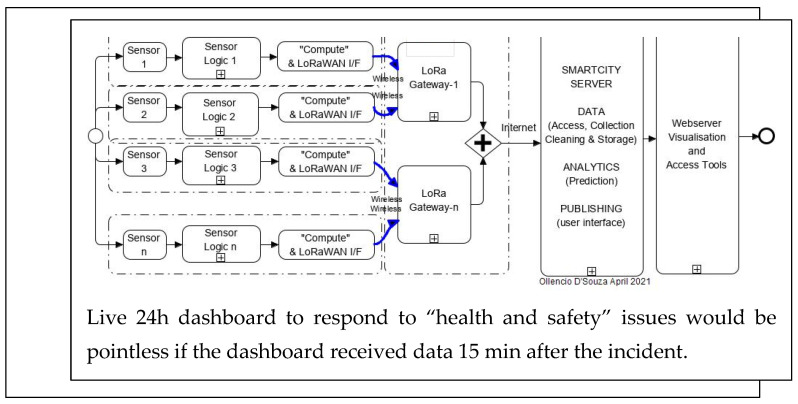
Schematic drawing shows the path of sensor data through infrastructure which could result in loss of temporal sync.

**Figure 32 sensors-22-08143-f032:**
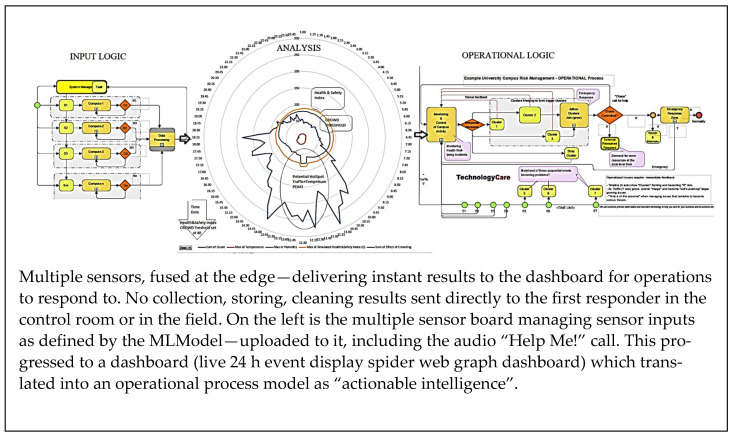
The fusion dashboard operational mode.

**Table 1 sensors-22-08143-t001:** The features and performance of a sensor fusion operational environment.

SNo	Operational Requirements	Desired Performance (Output)
1.	Embedded device for sensor fusion, real-time multiple sensor data analysis and results in storage	TinyML edge analytics ML model optimised for sensor data fusion and edge analytics. Only temporally synchronous application data will be stored so “prediction” and “detection” are achieved reliably by analysis.
2.	Power supply optimised for extended operational periods	“Wake” triggers conserve power. Renewable energy source, reliability monitoring and energy harvesting at the edge.
3.	Operating firmware reliability	Error checked for OTA (over the air) model updates.
4.	Reliable results transfer (upload)	Error checked, the optimised payload for transmission of analysed and computed results.

**Table 2 sensors-22-08143-t002:** A tabular listing of typical “wake” trigger options in each activity domain (truncated to fit).

Standard Multi-Domain “Wake” Trigger Options	Typical Risk Factor—Health	Security	Typical Risk Factor—Safety	Typical Risk Factor—Fire
Detecting a human or humans entering and leaving the sensor area in daylight	Several “sterile” areas could be “compromised” if people without PPE are allowed to roam unchallenged.	Designated “no go” (sterile) security areas are typical risk triggers set up to keep “intruders” away.	Building areas with cranes, etc., must be isolated so no one gets hurt. Thus, a man or machine “wake triggers” applied.	Often a “roll call” is taken on a fire alert to see who is on the premises. Any human or animal presence triggers “wake triggers”.
Detecting a human or humans entering and leaving the sensor area in “darkness” (no light)	Dark areas have severe risk implications. Special sensors with IR illumination or high gain deliver similar situational awareness as in the day.	Dark areas have severe risk implications, but illumination triggered by human movement helps.	Dark areas have severe risk implications, and illumination is essential for safety. Lights could be automatically turned on.	Dark areas have severe risk/rescue implications. Smoke and carbon monoxide are the most significant threats—smoke/CO sensors are used to trigger the siren and lights.
Establishing the time of day	All incoming and outgoing personnel must be timed in and out.	All incoming and outgoing personnel must be timed in and out,	All incoming and outgoing personnel must be timed in and out.	All incoming and outgoing personnel must be timed in and out.
Establishing the route	Movement history	Movement history	Movement history	Movement history
Establishing the count for 24 h	Counts of people through to waiting rooms could be used to control congestion.	Counts determine traffic flow into service areas.	Counts are used to restrict people’s traffic to stop overloading.	Counts are used during the evacuation of people to safety.
Data generation failure	Fused sensor detectors deliver continuity.	Fused sensor detectors offer continuity.	Fused sensor detectors offer continuity.	Fused sensor detectors provide continuity.
Establishing “door open/closed/ajar	Essential to establish the status of the door to stop the spread of infections.	Door status is vital to isolating the threat.	Door status is essential to isolating threats by sealing the area.	Fire door status is very critical to establishing or maintaining safety.
Establishing bed movement and positioning	Specific to “aged care”. Location of crucial caregiving assets.	Unusual activity of assets in working space related to security.	Assets left unattended block vital thoroughfare.	Legislated fire safety requirements supervised autonomously.
Establishing staff attendance	Specific to “aged care”. Colour of staff clothing (see image Figure 6, blue).	Use of HiVisibility vests or uniforms (see image Figure 6, yellow).	Use of high-visibility and safety gear (see image Figure 6, yellow).	All fire kits are a specific colour (yellow/red), and hence can be easily recognised.

## Data Availability

Data both real and simulated was used in the trial and results presented in the paper. The study did not generate any extensive report data. The results were presented in a “Confusion Matrix Table”.

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
