# Peer review of "Health, Security and Fire Safety Process Optimisation Using Intelligence at the Edge"

_sensors, 2022, doi:10.3390/s22218143_

Round 1

Reviewer 1 Report

Overall, the paper presents an interesting application with what appears to be a practical deployment. There are also great diagrams for illustration. But the paper also needs  a substantial "Related Work" section to outline the contributions and novelty of the paper with respect to related work, and why the work warrants publication.

Also, the paper needs serious proof-reading and updates if it is to be published, some comments below, e.g.:

-line 2 title: "Securityand" and "FireSafety"?

-line 45: "Key Issues:"? Perhaps put this into a proper sentence?

-line 48: I think it is hard to generalise this 68% to say this is an issue since it would be application specific - need to say more about how [6] made such a conclusion

-line 108, "!" in "...dashboard!" not required?

-line 120, "Table.1"? Just "Table 1" will do - check throughout the paper please

-line 127 "havea"?

-try to put a space before a reference, e.g., line 115: instead of "...edge[21]" use "...edge [21]"

-section starting on Line 125, explain carefully what the domain and scenario are, e.g., is it a building, a room, how many people typically involved, size of area etc? And what is to be detected and why?

-section starting on line 169: please provide more details of the MCUs used: brand, model, features etc, what type of sensors are they, and their capabilities etc; and how they have been used? Where are they positioned and why? Give a diagram - perhaps layout of building / floor showing position of sensors etc...

-paragraph starting on line 237, please elaborate - idea that no so clear?

-the TinyMLmodel is mentioned at various places: please explain TinyML earlier providing background on it and it is mentioned in Figure 10 CNN is used - please provide details of the CNN used with TinyML

-section start on line 276: looks more like a draft with bullet points; please rewrite and elaborate - if the steps are sequence, number them....

-in Figure 14 caption, mission double quotes ..."...a radio crakcle" and misspelling on "crakcle"?

-section starting line 336: Again please write into proper paragraphs - e.g., "Calling for help..." sounds like dot points which need to be expanded into proper sentences and paragraphs; please check other sections

-line 467, there is mention of scenarios etc, but the description of these scenarios seem inadequate?

Author Response

Separate file attached

Reviewer 2 Report

The content of this paper was to introduce the application of the edge. It provides novel theories and new techniques. It is worth recommending as a research paper.  However, the way of expression needs to be enhanced to make the paper to be easy to read.

What is the scope of this study? In the title, they are “Health, Security and Fire Safety Process” and “a solution for the health, security, safety, and fire domains” in the abstract.

Please rewrote the paper. First, the novel theories and the flow chart of these new techniques were described in detail in the section on Method and Materials. Then some actual cases, such as the content in figure 2, figure 6, figure 11, figure 14, figures 15 -19, etc. were described according to the flow chart of the techniques.

If the readability is improved, it will be an excellent paper.

Author Response

Separate file attached

Reviewer 3 Report

Report on paper "Health, Security and Fire Safety Process Optimisation Using Intelligence at the Edge" submitted by D'Souza et al., for publication in Sensors (sensors-1934331).

The authors developed a multi-sensor fusion framework with energy conservation and investigated optimization techniques using anomaly detection modes to deliver real-time insights in demanding life-saving situations. While the paper is interesting, it cannot be accepted in its present form and the authors must perform some modifications by addressing the following comments:

  1. The originality of the paper should be clearly highlighted at the end of the introduction with respect to the state of the art.
  2. The content is globally very technical and the authors should place some details in an appendix without altering the comprehension of the paper.
  3. At the end of section 4, the limitations of the developed approach should be discussed from a critical point of view.
  4. Is it possible to propose some quantitate indicators to highlight the efficiency of the proposed approach? 
  5. The quality of several figures should be enhanced.

Author Response

Separate file is attached

Round 2

Reviewer 1 Report

The paper definitely has improved, but a number of presentation issues still.

Lines 89 to 100 argues for the originality and novelty of the work but without a substantial literature review of (and comparison with) related work and systems, appropriate to the application at hand, one cannot accept these claims - in the literature, what have been similar systems deployed for research or (where documented) in industry? What design principles have been employed, similar or different from related work? What other similar applications have there been, similar sensor deployments etc? What is unique about the work in this paper compared to these other systems? etc ... There are many smart city sensor networks and deployments and no others acknowledged or discussed? What design rationale and ideas are similar to those in other systems in the literature? Without such discussion, this is not a research paper but reads like a project report, which is not publishable in a research journal. There is now a "Related work -..." section but this is mainly describing the context of the research, not related work from the research literature, and deeper discussions and comparisons. Please add a major section "2. Related Work" which could come after the "1. Introduction"...

The Introduction and Conclusion also needs to highlight the main research contributions of the paper compared to existing work in the research literature. For example, in the Conclusion, what are lessons learnt? How will the research community build systems now, given what you have done? What are the design principles illustrated? 

Figure 1 and Figure 2 are a little too blur - the words on the graphs can hardly be read? Similarly with words in Figures 3, 4 and 5 ...as well as Figures 7,8,9, 14, 15,16,17, 19,22,23,24,25,26,27,31,32 (Figure 32 is too small for the content given - also check other figures as well which might need to be enlarged for reusability) ... Figure 19 has two images, perhaps call it 19(a) and 19(b), and Figure 20 should be to the left of Figure 21...  etc please double check all figures that the words are easy to read and clear.

And no need to use exclamation marks (as used in a number of places)... e.g., "...time!." etc. Still needs proof reading - just in the first line of the abstract "The Proliferation of..."  should be "the proliferation of..." - please throughout the paper again...

Author Response

Open Review (Reviewer 1 – second round)

English language and style

( ) Extensive editing of English language and style required
(x) Moderate English changes required
( ) English language and style are fine/minor spell check required
( ) I don't feel qualified to judge about the English language and style

Yes

Can be improved

Must be improved

Not applicable

Does the introduction provide sufficient background and include all relevant references?

( )

(x)

( )

( )

Are all the cited references relevant to the research?

(x)

( )

( )

( )

Is the research design appropriate?

(x)

( )

( )

( )

Are the methods adequately described?

( )

(x)

( )

( )

Are the results clearly presented?

( )

(x)

( )

( )

Are the conclusions supported by the results?

( )

(x)

( )

( )

Comments and Suggestions for Authors

The paper definitely has improved, but a number of presentation issues still.

Lines 89 to 100 argues for the originality and novelty of the work but without a substantial literature review of (and comparison with) related work and systems, appropriate to the application at hand, one cannot accept these claims - in the literature, what have been similar systems deployed for research or (where documented) in industry?

(Reviewer extended remarks after this response)

Reviewer-1, Second round of comments

Author Response

The paper definitely has improved, but a number of presentation issues still.

Thank you.  Will review your comments and respond.

Lines 89 to 100 argues for the originality and novelty of the work but without a substantial literature review of (and comparison with) related work and systems, appropriate to the application at hand, one cannot accept these claims - in the literature, what have been similar systems deployed for research or (where documented) in industry?

I do not understand the premise of this statement and requirement because I have constructed this original thought on collateral, collaboration, corroboration and reinforcement.  If this is an original thought how can there be any reference to it anywhere else?  It is based on the logic “created” in workflow drawings presented and examples of these logical executions presented in specific cases such as a door opening and pressure sensing.  I am unsure what references would be relevant because I have presented references to using TinyML and relevant work in other domains.

The two reference papers provided by reviewer 1 were not related to the subject of “real-time” life and death situation responses.  One on drones is “data collection, estimation and prediction” and the other paper is similar.

What design principles have been employed, similar or different from related work?

If “similar” work is required what is the purpose of a paper that presents new ideas on methods to deliver reliable un-truncated real time responses using “fused” sensor data?

The design principles were based on a problem presented in several respected papers on the “loss and delay” in data because of infrastructure issues, truncated data, False and unwanted event data and general data handling of huge amount of single sensor data at the receiving end.

We have proposed sensor fusion and analysis using TinyML models to overcome all these issues.

We also present a solution for “unwanted alarms” which reduces a waste of resources to respond making analysed information available directly to first responders.

Reference papers on these issues and describing attempts to manage data have been presented.

What other similar applications have there been, similar sensor deployments etc?

There have been many deployment but none based on the techniques expounded in my paper.

None of these have worked on data integrity issues in real time and temporal sync of data and real time responses.

What is unique about the work in this paper compared to these other systems? etc ...

As explained earlier;

Problems experienced in industry – both in errors in data and delays to data arriving from mostly singular sensors.

We have proposed the management of this by the following;

Edge intelligence to collect and analyse the data can be directly sent to the first responder.

I could not explain it any other way than to use the terms “collaborate, corroborate, reinforce” for real time “fusion” that generated results that enables real time response using Machine Learning techniques.

There are many smart city sensor networks and deployments and no others acknowledged or discussed?

Although other papers have tackled these issues one at a time, I have picked four domains, developed (data engineered) the training parameters (tables), trained the models with this data and developed a model to do edge analysis, energy harvesting with wake words to deliver reliable results over very low power BLE links.  

If we were discussing “smart city” issues we would refer to a range of deployments.  But this paper is about using machine learning to “fuse” sensor information to deliver real-time response to overcome infrastructure congestions and false “unwanted” alarm rates that plague ithe industry.

What design rationale and ideas are similar to those in other systems in the literature?

The only thing “similar” is using the proven technology TinyML.  This technology is used in my paper to solve real-time industry issues.  We have researched the application of machine learning which is improving daily on capability and applications – to solve multiple domain issues (industry problems).

Without such discussion, this is not a research paper but reads like a project report, which is not publishable in a research journal.

I believe I have raised something new, solving problems referenced in the literature reviewed and delivered in domains requiring real-time responses to safeguard life and property.

I did explain the problem we were trying to combat – then explained the issues we were trying to overcome, and then demonstrated the technology which delivers us the tools to have the proper outcomes in 4 similar domains!

We studied these domains, “data engineered” the critical training requirements in tables and then proceeded to trial the solution.

We explained new concepts like collaboration, corroboration and reinforcement to ensure results are verified before they are sent to a response team so they do not waste resources.

At every step, we have been unique in our application of TinyML as a tool to handle huge amounts of data from MCUs with up to 11 sensors on board, fusing it to deliver a better understanding (situational awareness) to deliver better “actionable intelligence”

There is now a "Related work -..." section but this is mainly describing the context of the research, not related work from the research literature, and deeper discussions and comparisons.  Please add a major section "2.  Related Work" which could come after the "1.  Introduction"...

Related work details the experiences of handling data that was not real-time,  the required infrastructure, including a detailed drawing.  From this experience and published papers, we researched the best methods to overcome issues encountered, such as latency and congestion, effects on the analysis for “health and safety”, etc

Our paper refers to unique methods to develop logic that humans do well and then use ML and the power of electronic processing and speed – to build many more associations as “taught” to the machine learning configuration “model” deployed.

This technique is modern and ensures that we explore all aspects of the problem using “computing” resources to provide us the most effective outcome.

The Introduction and Conclusion also needs to highlight the main research contributions of the paper compared to existing work in the research literature.  For example, in the Conclusion,

a)     what are lessons learnt?

b)     How will the research community build systems now, given what you have done?

c)     What are the design principles illustrated? 

a)     Lessons learnt?  We have presented problems in tables, related work with references for which we have built solutions – and successfully presented results and demonstrated it via a trial with results!.

b)     The research community will start using “Machine Learning” in many different ways.  Like we are doing with real-time “fusion” and response to events, overcoming “infrastructure issues” and reducing false alarms using collaboration, corroboration, and reinforcement techniques, which are unique!  These will reduce false alarms and make event response effective because of excellent “situational awareness” and efficient because of “Actionable Intelligence”!

c)     Logic drawings presented are based on decision tree logic, congestion management using Queuing theory, Confusion Matrix to “assess” outcomes and adopting newly expounded TinyML design principles!  These design principles have been implemented on a trial and results presented as part of this research.

Figure 1 and Figure 2 are a little too blur - the words on the graphs can hardly be read?

These figures have been improved.  The PDF is always grainier – so reviewing it in a word document would be ideal because the resolutions captured in word are much higher.  But we have made every effort to make all drawings bigger.

Similarly with words in Figures 3, 4 and 5 ...as well as Figures 7,8,9, 14, 15,16,17, 19,22,23,24,25,26,27,31,32 (Figure 32 is too small for the content given - also check other figures as well which might need to be enlarged for reusability) ...

We have increased the size of the drawing on the page to improve readability.  Also these “graphed” examples do not depict results – they are there as an explanation not proof!

I would request that you please read the word document.  The Microsoft print-to PDF does not retain the resolution of the word document as much as we would like.

Figure 19 has two images, perhaps call it 19(a) and 19(b), and

This numbering has been changed.

Figure 20 should be to the left of Figure 21...  etc please double check all figures that the words are easy to read and clear.

Noted: will be redone in cute pdf.  Images in word seem to be OK.

And no need to use exclamation marks (as used in a number of places)... e.g., "...time!." etc. Still needs proof reading - just in the first line of the abstract "The Proliferation of..."  should be "the proliferation of..." - please throughout the paper again...

Exclamation marks will be removed.

Energy-Efficient Inference on the Edge Exploiting TinyML Capabilities for UAVs. Drones 20215, 127.  doi: 10.3390/drones5040127

The reference suggested is not meant for “real-time domains” and uses longer periods to collect data to “predict” (a period of data collection is required before prediction can be done)

 Machine Learning Approach Using MLP and SVM Algorithms for the Fault Prediction of a Centrifugal Pump in the Oil and Gas Industry. Sustainability 202012, 4776. doi: 10.3390/su12114776

This reference is also very different to the domain requirements discussed in our research – which required “real-time” response and hence real-time “fused” data – using collaboration, corroboration, collateral and reinforcement.

What design principles have been employed, similar or different from related work?

What other similar applications have there been, similar sensor deployments etc?

What is unique about the work in this paper compared to these other systems? etc ...

There are many smart city sensor networks and deployments and no others acknowledged or discussed?

What design rationale and ideas are similar to those in other systems in the literature?

Without such discussion, this is not a research paper but reads like a project report, which is not publishable in a research journal.

There is now a "Related work -..." section but this is mainly describing the context of the research, not related work from the research literature, and deeper discussions and comparisons.  Please add a major section "2.  Related Work" which could come after the "1.  Introduction"...

The Introduction and Conclusion also needs to highlight the main research contributions of the paper compared to existing work in the research literature.  For example, in the Conclusion,

  1. what are lessons learnt?
  2. How will the research community build systems now, given what you have done?
  3. What are the design principles illustrated? 

Figure 1 and Figure 2 are a little too blur - the words on the graphs can hardly be read? 

Similarly with words in Figures 3, 4 and 5 ...as well as Figures 7,8,9, 14, 15,16,17, 19,22,23,24,25,26,27,31,32 (Figure 32 is too small for the content given - also check other figures as well which might need to be enlarged for reusability) ...

Figure 19 has two images, perhaps call it 19(a) and 19(b), and

Figure 20 should be to the left of Figure 21...  etc please double check all figures that the words are easy to read and clear.

And no need to use exclamation marks (as used in a number of places)... e.g., "...time!." etc. Still needs proof reading - just in the first line of the abstract "The Proliferation of..."  should be "the proliferation of..." - please throughout the paper again...

Submission Date

09 September 2022

Date of this review

06 Oct 2022 03:01:04

Open Review – Reviewer 2 – second round

English language and style

( ) Extensive editing of English language and style required
( ) Moderate English changes required
(x) English language and style are fine/minor spell check required
( ) I don't feel qualified to judge about the English language and style

Yes

Can be improved

Must be improved

Not applicable

Does the introduction provide sufficient background and include all relevant references?

( )

(x)

( )

( )

Are all the cited references relevant to the research?

( )

(x)

( )

( )

Is the research design appropriate?

( )

(x)

( )

( )

Are the methods adequately described?

( )

(x)

( )

( )

Are the results clearly presented?

( )

(x)

( )

( )

Are the conclusions supported by the results?

( )

(x)

( )

( )

Comments and Suggestions for Authors

The revised paper has been improved significantly.  All commend are replied.

Submission Date

09 September 2022

Date of this review

06 Oct 2022 02:46:44

Reviewer 2 Report

The revised paper has been improved significantly. All commend are replied.

Author Response

Thank you for your comments. We have made every effort to meet all suggested updates.

Reviewer 3 Report

In several figures, the text is very small (almost unreadable). The authors must increase the text size and enhance the resolution to meet the journal requirements. 

Author Response

Open Review

English language and style

( ) Extensive editing of English language and style required
(x) Moderate English changes required
( ) English language and style are fine/minor spell check required
( ) I don't feel qualified to judge about the English language and style

Yes

Can be improved

Must be improved

Not applicable

Does the introduction provide sufficient background and include all relevant references?

( )

(x)

( )

( )

Are all the cited references relevant to the research?

( )

(x)

( )

( )

Is the research design appropriate?

( )

(x)

( )

( )

Are the methods adequately described?

( )

(x)

( )

( )

Are the results clearly presented?

( )

(x)

( )

( )

Are the conclusions supported by the results?

( )

(x)

( )

( )

Comments and Suggestions for Authors

In several figures, the text is very small (almost unreadable). The authors must increase the text size and enhance the resolution to meet the journal requirements. 

Response from Authors:

1 In several figures, the text is very small (almost unreadable)

All figures have been enlarged and improved – unfortunately going to a PDF process does not retain the original resolution of the images especially if they are very complex.

However none of the images are “results” and are there to explain the application.  The current resolution should be adequate to support the reason for inclusion such as “’Related Work”

  1. The authors must increase the text size and enhance the resolution to meet the journal requirements. 

All efforts have been made to meet journal requirements fro relevant data that is required to meet the results display.
